# Comparative Toxicological Effects of Insecticides and Their Mixtures on *Spodoptera littoralis* (Lepidoptera: Noctuidae)

**DOI:** 10.3390/insects16080821

**Published:** 2025-08-07

**Authors:** Marwa A. El-Saleh, Ali A. Aioub, El-Sayed A. El-Sheikh, Wahied M. H. Desuky, Lamya Ahmed Alkeridis, Laila A. Al-Shuraym, Marwa M. A. Farag, Samy Sayed, Ahmed A. A. Aioub, Ibrahim A. Hamed

**Affiliations:** 1Plant Protection Research Institute, Agricultural Research Center, Dokki 12618, Giza, Egypt; marwa_saleh2001@yahoo.com (M.A.E.-S.); wahieddesuky_1974@yahoo.com (W.M.H.D.); 2Department of Plant Protection, Faculty of Agriculture, Zagazig University, Zagazig 44511, Sharkia, Egypt; aliaioub1965@yahoo.com (A.A.A.); eaelsheikh@zu.edu.eg (E.-S.A.E.-S.); ibrahim_hamed2002@yahoo.com (I.A.H.); 3Department of Biology, College of Science, Princess Nourah Bint Abdulrahman University, P.O. Box 84428, Riyadh 11671, Saudi Arabia; laalkeridis@pnu.edu.sa (L.A.A.); laalshuraym@pnu.edu.sa (L.A.A.-S.); 4Department of Economic Entomology and Pesticides, Faculty of Agriculture, Cairo University, Al Giza 12613, Giza, Egypt; marwa_farag82@agr.cu.edu.eg

**Keywords:** toxicology assessment, insecticide mixtures, insect resistance, detoxification enzymes, *Spodoptera littoralis*

## Abstract

This study evaluated the effectiveness of different insecticides and their combinations in controlling *Spodoptera littoralis* under field conditions. The insecticides tested included emamectin benzoate, lufenuron, cypermethrin, chlorpyrifos, and spinosad. The results showed that emamectin benzoate was the most effective, with over 80% effectiveness, followed by lufenuron. Spinosad had the lowest effectiveness. Insecticide combinations were also tested; it was found that the combination of emamectin benzoate and lufenuron was the most effective, achieving over 90% effectiveness, while emamectin benzoate and spinosad were the least effective. However, mixing insecticides led to higher pesticide use and lower control effectiveness compared to individual applications. The study also found that different insecticide combinations affected enzyme activities, such as alfa-esterase, beta-esterase, and acetylcholinesterase, in larvae. Using insecticides individually was more economically viable than using mixtures for controlling cotton leafworm in cotton crops.

## 1. Introduction

The cotton leafworm, *Spodoptera littoralis* (Boisd.) (Lepidoptera: Noctuidae), is a major insect pest in Egypt, inflicting significant damage to a variety of crops and vegetables, particularly cotton [1]. The larvae of *S. littoralis* are highly polyphagous, leading to significant economic damage in both open fields and greenhouses across a wide range of ornamental, industrial, and vegetable crops. Furthermore, many populations have developed resistance to most conventional insecticide groups [2], highlighting the need to introduce new insecticides that are effective, safer for humans, and have minimal impact on the ecosystem. In the field, insecticide mixtures are commonly employed to broaden the range of control when multiple pests are present at the same time. Insecticide mixtures are also advised to improve control effectiveness against a single pest, delay the development of insecticide resistance, or address existing resistance in a pest species. The use of mixtures as a strategy for managing resistance in insect pests has been recommended by numerous researchers [3]. Mixtures are either supplied as pre-mixed formulations by pesticide companies or prepared by farmers through tank mixing. In an ideal scenario, insecticides with different modes of action are combined, assuming they work synergistically to eliminate the target pest. However, the effects of mixing can vary across various insect species or strains, depending on their physiology and the resistance mechanisms they have developed. Spinosad is an insecticide produced through fermentation of the naturally occurring soil actinomycete, *Saccharopolyspora spinosa* [4]. Spinosad works in two distinct ways by targeting nicotinic acetylcholine receptors and gamma-aminobutyric acid (GABA) receptors [5]. Emamectin benzoate (EB) is a second-generation analog of avermectin, known for its outstanding effectiveness against lepidopteran pests [6]. EB is a novel semisynthetic derivative of the natural product abamectin, belonging to the avermectin family. It acts by binding to glutamate-gated chloride channels in the insect’s nervous system, causing prolonged channel opening, chloride ion influx, hyperpolarization, and ultimately paralysis, leading to the insect’s death [7]. Cypermethrin (CYP) is a synthetic pyrethroid that is widely utilized as an insecticide in crop protection to minimize yield loss and maximize yield quality in numerous countries [8]. CYP is a modified derivative of pyrethrins, which are natural substances extracted from the flowers of *Chrysanthemum cinerariaefolium* [9] and are known to affect the nervous system, specifically by deleting the closure of sodium channels, causing repetitive after-discharges that result in hyperexcitation of the nervous system [10]. CYP has low mammalian toxicity [11] and highly effective insecticidal properties against insects from several orders, including Coleoptera, Diptera, Hemiptera, and Lepidoptera [12]. Chlorpyrifos is an organophosphate insecticide that kills insects by inhibiting acetylcholinesterase, an essential enzyme for nerve function, leading to overstimulation of the nervous system and paralysis [13]. Although its use has been banned or restricted in many countries due to health and environmental concerns, it is still commonly used in Egypt, especially in cotton production [14]. Insect growth regulators (IGRs) encompass juvenile hormone (JH) analogs and chitin synthesis inhibitors (CSIs).

Chitin synthesis inhibitors (CSIs), like lufenuron, block the production of chitin, a key component of the insect exoskeleton. Insects treated with CSIs are unable to produce a new cuticle, preventing them from successful molting into the next developmental stage [15]. The effectiveness of insecticides and their combinations with IGRs against the cotton leafworm has drawn the attention of numerous researchers [16]. Lufenuron affects the development of lepidopteran larvae and leads to the production of infertile eggs. Insects treated with it develop normally until molting, at which point they are unable to complete the process due to the inhibition of new cuticle synthesis [17]. Due to the significant role of insecticide mixtures in mitigating insect resistance, several compound mixtures have been introduced to the market in Egypt. Therefore, it is relevant to investigate the effects of certain new insecticide mixtures on *S. littoralis*. In lepidopteran populations, insecticide resistance primarily involves two mechanisms: enhanced detoxification and target site insensitivity. The activity of detoxifying enzymes is necessary for an insect to survive toxic surroundings such as insecticides [18]. The process of cellular detoxification in insects can be separated into three phases: I, II (involving metabolizing enzymes), and III (involving transporters) [19]. Cytochrome P450 monooxygenase, glutathione S-transferase (GST), and carboxylesterase (CarE) are the primary enzymes involved in phases I and II, whereas phase III is dominated by ATP-binding cassette (ABC) transporters [20]. Our study aimed to investigate the comparative toxicity of the recommended insecticides emamectin benzoate, spinosad, lufenuron, cypermethrin, and chlorpyrifos and their binary mixtures on *S. littoralis* under both laboratory and field conditions. This study also examined the biological and biochemical effects of these insecticides on enzyme activity in both field and laboratory strains.

## 2. Materials and Methods

### 2.1. Insecticides and Chemicals

The tested insecticides were emamectin benzoate (Speedo^®^ 5.7% WDG, Hebei Veyong Bio-chemical Co., Ltd., Shijiazhuang, China), spinosad (Tracer^®^ 24% SC, Dow Agro Sciences, Abingdon, UK), lufenuron (Match^®^ 5% EC, Syngenta Agro, Basel, Switzerland), chlorpyrifos (Chlorfan^®^ 48% EC, KafrElzayate, Gharbia, Egypt), and cypermethrin (Fury^®^ 10% EW, FMC Chemical S.P.R.I, Woluwe-Saint-Lambert, Belgium). Chemicals that were used in the determination of acetylcholine esterase were purchased from Sigma-Aldrich Chemical Ltd., Cairo, Egypt. Reagents that were used in the determination of the activity of other enzymes were obtained from Bio-Diagnostic Company, Dokki, Egypt.

### 2.2. Test Insects

The laboratory reference strain (L-strain) of *S. littoralis* was originally obtained from the Plant Protection Institute, ARC, Giza, Egypt. This strain had never been exposed to insecticides before. Insects were reared under laboratory conditions at 27 ± 2 °C, 65 ± 5% relative humidity, and a 16:8 (L:D) photoperiod for six consecutive generations, as described by [21]. Hatched egg masses were placed in 500 mL glass jars covered with muslin cloth and provided daily with leaves of fresh castor bean (*Ricinus communis*) as food until pupation. The pupae were moved to glass jars, including filter papers, and kept in cages (35 × 35 × 35 cm) until the emergence of adults. Newly emerged moths were sexed and kept in pairs in clean 1 L capacity jars. Each jar was supplied with cotton wool soaked in a 10% honeybee solution for feeding, which was replaced daily to prevent fermentation and microbial growth. For egg laying, fresh green leaves of Nerium oleander were provided.

The field strain (F-strain) of *S. littoralis* used in this investigation was collected from a cotton field in Zagazig, Egypt, during the 2023 and 2024 seasons. It was then reared for one generation with the same laboratory conditions as the L-strain.

### 2.3. Bioassay

Bioassays were conducted on the second- and fourth-instar larvae of both the F-strain and L-strain to compare the effectiveness of the tested insecticides. A range of aqueous concentrations (0.25, 0.5, 1, 2, 4, 8, and 16 mg/kg) of each insecticide was prepared. The leaf-dipping bioassay method was employed to determine the LC_25_, LC_50_, LC_90_, and slope, according to El-Ghar [22]. The treated leaves were offered to newly molted second- and fourth-instar larvae from both strains. Dry and clean castor bean leaves (0.5 cm × 0.5 cm) were immersed for 30 s in an insecticide solution of a given concentration, air-dried for 30 min at room temperature, and then given to the larvae in a glass jar covered with muslin for 24 h. Ten larvae were used for each concentration, with three replicates per concentration, totaling 30 larvae per concentration. Leaves dipped in distilled water served as the control group. Data were collected after three days according to Finney [23]. Each treatment was repeated three times, and larvae were considered dead if no movement was detected when touched with a fine brush. Corrected mortality was estimated using the formula of Abbott [24]. The resistance ratio (RR) was determined by dividing the LC_50_ value of the F-strain by that of the L-strain. The toxicity index (TI) was calculated using [25], by dividing the LC_50_ or LC_90_ of the most effective compound by that of the other compound, multiplied by 100. Relative potency values were calculated according to the method described by Zidan and Abdel-Megeed [26], who divided the LC_50_ or LC_90_ of the less effective compound by that of the more effective compound.

### 2.4. Biological Parameters

Prewashed clean leaves of castor bean were dipped for 30 s in the concentrations of lufenuron, cypermethrin, spinosad, or chlorpyrifos in binary mixtures. Ten second-instar larvae of *S. littoralis* were transferred into a glass jar (350 mL) and provided with treated castor bean leaves. Four replicates of 10 larvae (one-day-old) of the second instar were used, so a total of 40 larvae were used for each insecticide and fed for 24 h. Leaves were dipped in a water-only control. The surviving larvae were placed in clean jars and given untreated leaves until pupation. For mating experiments, the emerged moths, whether from treated or untreated control larvae, were sexed, placed in glass jars, and provided with Nerium oleander leaves as an oviposition site. A piece of cotton soaked in a 10% sugar solution was offered daily for feeding. The eggs were counted and placed in a clean jar with untreated castor bean leaves until hatching. Newly hatched larvae were recorded to calculate the hatchability percentage, weight of pupae (mg), and percentages of adult emergence. Two male adults and one female adult from each replicate were placed in a clean jar to maximize the probability of successful mating. The longevity of the adult stage, fecundity, incubation period, and percentages of egg hatching were also recorded.

### 2.5. Field Experiment Procedure

The field experiments were carried out during the 2023 and 2024 cotton seasons, in the Zagazig region, El-Sharkia Governorate, in an area of 1 feddan (fed.) (4200 m^2^). All treatments were assigned to plots (105 m^2^) in a randomized complete block design (RCBD) cultivated with cotton plants, *Gossypium barbadense* L., variety Giza 94. The cotton growing season extended from 20 March to the end of October during the two seasons, with temperatures ranging from 27 to 35 °C, a relative humidity between 45 and 65%, and minimal rainfall. The experimental site had clay loam soil with a pH of 7.6, 1.9% organic matter, and moderate levels of essential nutrients, ensuring uniform soil fertility. The tested compounds were applied with the recommended rates on 20 June and 24 June during the 2023 season and on 19 June and 22 June during the 2024 season. All agricultural practices were carried out according to recommendations. Treatments were performed with individual insecticides of emamectin benzoate (80 g/fed.), lufenuron (160 mL/100 L), chlorpyrifos (1 L/fed.), cypermethrin (250 mL/fed.), and spinosad (30 mL/100 L). The binary mixtures of these insecticides with half the recommended field rates of emamectin benzoate were tested with the field rate of other insecticides. The field was divided into ten treatments; five were treated with the tested insecticides, and four treatments were treated with mixtures of these insecticides, while ten treatments served as a control. Four replicates were set for each treatment. Twenty-five cotton plants were selected for each replicate, and all instar larvae were counted and recorded just before application and after 1, 7, and 10 days for all insecticides except lufenuron (3, 7, and 10 days) according to the insecticide evaluation protocol. The control treatment was sprayed with water only. Insecticide applications on cultivated cotton plants were achieved using knapsack sprayer equipment (CP3) at a rate of 200 L per feddan. Reduction percentages in insect number in cotton or potato treatments were calculated according to the formula of Henderson and Tilton [27].

### 2.6. Enzyme Preparation

The fourth-instar larvae were treated with the LC_50_ of insecticides and their combinations, using 30 larvae in three replicates. Five surviving larvae (weighing 15–20 mg) from both the laboratory and wild strains were homogenized in 0.1 M ice-cold phosphate buffer (pH 6.7). The samples were then centrifuged at 4 °C at 10,000 rpm for 20 min. The homogenate was filtered through glass wool to remove the fat body and stored at −20 °C to be used as an enzyme source.

#### 2.6.1. General Esterase Activity Assay

General esterase activity was estimated according to Mohan and Gujar [28] by using α-NA or β-NA as substrates. The reaction mixture was as follows: 5 mL of substrate solution (3 × 10^−4^ M α-NA or β-NA, 0.1 M phosphate buffer, 1% acetone (PH 7.6), and 30 µL of larval homogenate. The mixture was incubated at 27 °C for 15 min, and then 1 mL of diazo blue color reagent (1% diazoblue B and 5% sodium lauryl sulfate in a ratio of 1:5, respectively) was added. The developed color was determined at 555 or 600 ηm. A microplate reader (Infinite M Nano, TECAN, Grödig, Austria) was used for the measurement of α- and β-naphthol produced from hydrolysis of the substrate, respectively.

#### 2.6.2. Carboxylesterase Assay

Carboxylesterase activity was estimated using the method of Mohan and Gujar [28]. A 100 µL enzyme solution from treated larvae was added to test tubes containing 100 µL of 0.3 mM α-naphthyl or β-naphthyl acetate as a substrate, along with 4.8 mL of 40 mM phosphate buffer (pH 6.8). Each tube was incubated in the dark at room temperature for 20 min. After gentle shaking, 1 mL of staining solution (1% fast blue BB salt in phosphate buffer (40 mM, pH 6.8) with 5% sodium dodecyl sulfate (SDS)) was added to each tube and incubated for 30 min at 20 °C. Absorbance was measured at 590 nm using the microplate reader (Infinite M Nano, TECAN, Grödig, Austria).

#### 2.6.3. Acetylcholinesterase Assay

Acetylcholinesterase (AChE) activity was assessed using the substrate of acetylcholine-iodide, following the method of Ellman et al. [29]. Two hundred microliters of enzyme stock solution, 100 µL of 0.075 M acetylthiocholine iodide, and 240 µL of 0.1 M phosphate buffer (pH 7.4) were combined and incubated for 15 min at 27 °C. Afterward, 500 µL of 0.1 M eserine was added and mixed. The change in absorbance was recorded at 412 nm using a microplate reader (Infinite M Nano, TECAN, Grödig, Austria).

#### 2.6.4. Glutathione S-Transferase

Glutathione S-transferase (GST) activity was determined according to Habig et al. [30]. First, 50 µL of 50 mM chlorodinitrobenzene (CDNB) and 150 µL of reduced glutathione (GSH) were added to 2.79 mL of 40 mM phosphate buffer (pH 6.8). Then, 10 µL of enzyme stock solution was added to the mixture, which was gently shaken and incubated for 2–3 min at 20 °C. The mixture was transferred to the sample cuvette of a UV spectrophotometer, and the change in absorbance was recorded at 340 nm for up to 5 min using a microplate reader (Infinite M Nano, TECAN, Grödig, Austria).

#### 2.6.5. Total Protein Assay

The concentration of total protein was estimated using Bonjoch and Tamayo [31]. The protein reagent was prepared by dissolving 100 mg of Coomassie Brilliant Blue G-250 in 50 mL of 95% ethanol. To this, 100 mL of 85% (*w*/*v*) phosphoric acid was added, and the solution was diluted to a final volume of 1 L. A sample solution (50 µL) or 50 µL of serial concentrations ranging from 10 to 100 µg of bovine serum albumin (BSA) for the standard curve was pipetted into test tubes. The volume in each test tube was adjusted to 1 mL with phosphate buffer (0.1 M, pH 6.7). An amount of 5 mL of protein reagent was then added, and the contents were vortexed. The absorbance at 595 nm was measured using a microplate reader (Infinite M Nano, TECAN, Grödig, Austria) after 2 min and before 1 h, with the blank being prepared by mixing 1 mL of phosphate buffer with 5 mL of protein reagent.

### 2.7. Statistical Analysis

Insect mortality parameters from the bioassay were estimated using the Probit Analysis software program v 3.1 [23], with corrections for control mortality. Biochemical data were analyzed using SPSS version 14.0 for Windows. The results are presented as arithmetic mean values with SD. One-way analysis of variance (ANOVA) was conducted, and differences between groups were determined using the Duncan test. The least significant difference (LSD) was applied in the ANOVA at *p* < 0.05.

## 3. Results

### 3.1. The Toxicity of the Investigated Insecticides Against S. littoralis

The presented data in Table 1 show the LC_50_, LC_90_, slope values, relative potency, and toxicity index values for the tested compounds against the second- and fourth-instar larvae of laboratory and field strains of *S. littoralis*, under laboratory conditions using the leaf-dipping method after 3 days of treatments. At the LC_50_ level, emamectin benzoate was the most toxic compound for the two tested second- and fourth-instar larvae of *S. littoralis* on both field and laboratory strains. The data obtained showed that emamectin benzoate was the most effective insecticide while spinosad showed the lowest toxic effect on the second- and fourth-instar larvae, respectively, in both the L-strain and F-strain. The results of the slope values indicated that the insect population was variable in their susceptibility toward insecticides with the leaf-dip method. Slope values of regression lines revealed that the larvae of *S. littoralis* were more homogenous in their susceptibility to the tested insecticides. According to the estimated LC_50_ and LC_90_ for the two tested instars, it could be stated that the second instar had a higher level of susceptibility towards all the tested insecticides than the fourth instar. At both mentioned levels, emamectin benzoate was the most potent insecticide, while chlorpyrifos was the least effective one.

### 3.2. Effect of the Investigated Insecticides and Their Mixtures on Biological Parameters of S. littoralis

The presented data in Table 2 reveal the effect of feeding the second-instar larvae of *S. littoralis* leaves of castor bean treated with emamectin benzoate at the LC_25_ level and spinosad, cypermethrin, chlorpyrifos, and lufenuron at the LC_50_ level on the biological aspects of *S. littoralis***.** Larval duration, pupation (%), pupal duration, pupal weight, pupal mortality (%), emergence (%), longevity of adults, number of eggs/females, incubation period, hatchability (%), and fertility were investigated and recorded. The data presented in Table 2 indicate that all treatments increased the percentage of larval mortality: emamectin benzoate (at the LC_25_ level) causes 27.5% mortality, and lufenuron and spinosad (at the LC_50_ level) cause 50 and 45% mortality, respectively. Chlorpyrifos and cypermethrin caused 50 and 47.5% mortality, respectively. All treatments impacted the duration of the larval stage compared to the control, which lasted 19 ± 0.90 days. Emamectin benzoate (LC_25_) reduced this period to 16.79 ± 0.16 days, while chlorpyrifos, cypermethrin, spinosad, and lufenuron (LC_50_) resulted in durations of 18.6 ± 0.14, 18.67 ± 0.13, 19.55 ± 0.05, and 19.6 ± 0.14 days, respectively. Pupation percentage decreased compared with the control: emamectin benzoate achieved a value of 72.5% at the LC_25_ level, while chlorpyrifos, spinosad, cypermethrin, and lufenuron achieved values of 50, 45, 47.5, and 50%, respectively. The mean of pupation decreased compared with the control in all treatments. The results showed that there was a reduction in the emergence of adults. The lowest emergence percentage was recorded as 42.5% for spinosad, while the highest was 70% for emamectin benzoate.

The fecundity of the second-instar larvae fed on castor bean leaves treated with the tested insecticides was lower compared to the control. Lufenuron achieved the highest significant reduction, recording 625 ± 55.90 eggs/female, while females in the control laid an average of 1125 ± 82.91 eggs/female. There was a significant reduction in both fecundity and hatchability in all treatments on *S. littoralis* and the biological parameters.

### 3.3. Interactions Between the LC_25_ and LC_50_ of Tested Insecticides on the Mortality of Second- and Fourth-Instar Larvae

Our data in Table 3 show the joint action of emamectin benzoate at the LC_25_ level and its mixtures with chlorpyrifos, spinosad, cypermethrin, and lufenuron at the LC_50_ level against the second- and fourth-instar larvae of *S. littoralis*. Our data show an additive effect between emamectin benzoate with chlorpyrifos (LC_25_), lufenuron (LC_25_ and LC_50_), and cypermethrin (LC_25_) in the second- and fourth-instar larvae of *S. littoralis*. Notably, potentiation was observed in the interactions between emamectin benzoate with cypermethrin (LC_50_) and chlorpyrifos (LC_50_) in the second- and fourth-instar larvae of *S. littoralis*. However, antagonistic effects were recorded in the combination of emamectin benzoate with spinosad (LC_25_ and LC_50_) in the second- and fourth-instar larvae of *S. littoralis*.

### 3.4. Field Efficiency of the Tested Insecticides Against the Cotton Leaf Worm

Reduction percentages in the number of cotton leaf worm larvae on potatoes at different times of insecticide application in the Zagazig region, Sharkia, in the 2023 and 2024 seasons are presented in Table 4. Our results demonstrated the effectiveness of some insecticides against *S. littoralis* infestation during these seasons. All investigated insecticides at the recommended concentrations were found to reduce the infestation of *S. littoralis* on plants compared to the control. The initial effect was measured as the reduction percentages of *S. littoralis* larvae after the first day of treatment for emamectin benzoate, spinosad, cypermethrin, and chlorpyrifos after 3 days of treatment for lufenuron. The differences in the times of the initial effect are due to the differences in the mode of action of each insecticide, while the residual effect was measured after 7 and 10 days post treatment for all of them. In the 2023 season, the highest initial effects were recorded for lufenuron followed by emamectin benzoate, while the lowest was recorded for spinosad. The residual effect was highest in emamectin benzoate followed by lufenuron in the mixture of emamectin benzoate + chlorpyrifos, which recorded the highest reduction in initial and residual effects. The annual mean of reduction in emamectin benzoate in the 2024 season was recorded as 86.57, 85.33, 84.75, 80.73, and 81.70, respectively, and the residual effect was 89.91, 88.00, 86.19, 86.12, and 85.39, respectively, in 2023. In the second season (2024), it was 91.08, 89.72, 87.31, 87.82, and 86.75, respectively. In the 2023 season, with emamectin benzoate/cypermethrin, emamectin benzoate/chlorpyrifos, emamectin benzoate/spinosad, and emamectin benzoate/lufenuron, the annual mean was recorded as 90.47, 95.98, 93.51, and 92.28, respectively. In 2024, it was 91.19, 93.54, 92.76, and 89.17, respectively.

### 3.5. Biochemical Assay

The results for α-NA and β-NA in Table 5 show that, for the laboratory strain, larvae exposed to cypermethrin alone (157.05 µg α-naphthol/min/g protein) exhibited a significant increase in activity, while the minimal significant activity was recorded with lufenuron (155.32 µg α-naphthol/min/g protein) when used alone. In the insecticide mixtures, the most prominent significant activity was observed with emamectin benzoate + lufenuron (163.65 µg α-naphthol/min/g protein), while the least significant activity was recorded with emamectin benzoate + chlorpyrifos (143.67 µg α-naphthol/min/g protein). For the field strain, the greatest significant activity was recorded with cypermethrin (795.97 µg α-naphthol/min/g protein), while the minimal significant activity was observed with lufenuron (371.79 µg α-naphthol/min/g protein) when used alone. In the mixtures, the strongest significant activity was seen with emamectin benzoate + chlorpyrifos (485.49 µg α-naphthol/min/g protein), while the least significant activity was recorded with emamectin benzoate + lufenuron (215.81 µg α-naphthol/min/g protein).

For β-NA, the greatest significant activity in the laboratory strain was recorded with chlorpyrifos (135.16 µg product/min/g protein), while the lowest significant activity was recorded with spinosad (126.28 µg product/min/g protein) when used alone. When larvae were exposed to emamectin benzoate co-administered with lufenuron (127.32 µg product/min/g protein) or chlorpyrifos (118.60 µg product/min/g protein), a decrease in activity was observed. In the field strain, the most pronounced significant activity was recorded with spinosad (247.58 µg product/min/g protein), while the least significant activity was observed with cypermethrin (130.97 µg product/min/g protein) when used alone. When larvae were exposed to emamectin benzoate co-administered with chlorpyrifos (337.85 µg product/min/g protein) or spinosad (189.71 µg product/min/g protein), notable activity was also recorded.

For α-NA, larvae exposed to chlorpyrifos alone (36.40 µg α-naphthol/min/g protein) exhibited a significant increase in enzyme activity, while larvae treated with spinosad (22.30 µg α-naphthol/min/g protein) showed a significant decline in activity when the insecticides were used alone. In the insecticide mixtures, the strongest significant activity was observed with emamectin benzoate + chlorpyrifos (57.40 µg α-naphthol/min/g protein), while the least significant activity was recorded with emamectin benzoate + lufenuron (23.60 µg α-naphthol/min/g protein). For the field strain, the most significant activity was recorded with lufenuron (98.60 µg α-naphthol/min/g protein), while the least significant activity was observed with emamectin benzoate (49.40 µg α-naphthol/min/g protein) when used alone. In insecticide mixtures, the most pronounced significant activity was recorded with emamectin benzoate + spinosad (698.1 µg α-naphthol/min/g protein), while the least significant was found with emamectin benzoate + chlorpyrifos (319.5 µg α-naphthol/min/g protein).

For β-NA, in the laboratory strain, the greatest significant activity was recorded with spinosad (590.0 µg product/min/g protein), while the minimal activity was observed with cypermethrin (151.6 µg product/min/g protein) when the insecticides were used alone. When larvae were exposed to emamectin benzoate co-administered with spinosad (426.3 µg product/min/g protein), there was a substantial increase in activity, whereas co-administration with cypermethrin (129.60 µg product/min/g protein) caused a significant decrease in activity. In the field strain, the greatest significant activity was recorded with spinosad (884.7 µg product/min/g protein), while the least was recorded with cypermethrin (255.40 µg product/min/g protein) when used alone. When larvae were exposed to emamectin benzoate co-administered with spinosad (698.1 µg product/min/g protein), significant activity was observed, whereas the combination of emamectin benzoate and chlorpyrifos resulted in a co-toxicity factor of 319.50 µg product/min/g protein.

Lastly, for the α-NA, the greatest significant activity in the laboratory strain was recorded in larvae exposed to chlorpyrifos (10.76 µg α-naphthol/min/g protein), while the minimal significant activity was observed in larvae treated with lufenuron (863.0 µg α-naphthol/min/g protein) when insecticides were applied alone. In insecticide mixtures, the most pronounced significant activity was recorded with emamectin benzoate + chlorpyrifos (1037.8 µg α-naphthol/min/g protein), while the least significant activity was recorded with emamectin benzoate + spinosad (784.1 µg α-naphthol/min/g protein). For the field strain, larvae exposed to spinosad showed a notable increase in enzyme activity (1428.7 µg α-naphthol/min/g protein), while the least significant activity was recorded in larvae treated with lufenuron alone (1127.5 µg α-naphthol/min/g protein). Larvae exposed to the mixture of emamectin benzoate + cypermethrin exhibited a marked increase in enzyme activity (1298.4 µg α-naphthol/min/g protein), whereas the minimal significant activity was recorded in larvae treated with emamectin benzoate + spinosad (1038.4 µg α-naphthol/min/g protein). On the other hand, the AChE activity was recorded as 1037.8 and 1189.1 µg product/min/g protein in the L and F strains under EB + chlorpyrifos stress and 784.1 and 1038.4 µg α-naphthol/min/g protein under EB + spinosad stress, respectively.

Figure 1 shows that, for the laboratory strain, the highest significant activity was recorded in larvae exposed to chlorpyrifos (301.1 mmol-subconjugated min^−1^ g^−1^ protein), while the lowest significant activity was recorded in larvae treated with spinosad (262.3 mmol-subconjugated min^−1^ g^−1^ protein) when insecticides were used alone. In the insecticide mixtures, the highest significant activity was observed in larvae treated with emamectin benzoate + lufenuron (288.4 mmol-subconjugated min^−1^ g^−1^ protein), while the lowest significant activity was recorded in larvae treated with emamectin benzoate + cypermethrin (205.1 mmol-subconjugated min^−1^ g^−1^ protein). For the field strain, the highest significant activity was recorded in larvae treated with chlorpyrifos (437.9 mmol-subconjugated min^−1^ g^−1^ protein), while the lowest significant activity was observed in larvae treated with lufenuron (316.7 mmol-subconjugated min^−1^ g^−1^ protein) when used alone. In insecticide mixtures, the highest significant activity was recorded in larvae treated with emamectin benzoate + chlorpyrifos (462.6 mmol-subconjugated min^−1^ g^−1^ protein), while the lowest significant activity was recorded in larvae treated with emamectin benzoate + cypermethrin (344.5 mmol-subconjugated min^−1^ g^−1^ protein).

Figure 2 illustrates that, for the laboratory strain, the highest significant enzyme activity was observed in larvae treated with lufenuron (3.1 mg g^−1^ protein), while the lowest activity was recorded in larvae exposed to cypermethrin (2.1 mg g^−1^ protein) when used alone. Among the insecticide mixtures, the highest significant activity was seen in larvae treated with emamectin benzoate + spinosad (2.8 mg g^−1^ protein), while the lowest was observed in larvae treated with emamectin benzoate + cypermethrin and chlorpyrifos (2.3 mg g^−1^ protein). In the field strain, the highest significant activity was recorded in larvae treated with spinosad (5.7 mg g^−1^ protein), while the lowest activity was seen in larvae treated with cypermethrin (3.2 mg g^−1^ protein) when used alone. For the insecticide mixtures, the highest significant activity was observed in larvae treated with emamectin benzoate + spinosad (5.2 mg g^−1^ protein), whereas the lowest was recorded in larvae treated with emamectin benzoate + lufenuron (3.4 mg g^−1^ protein).

## 4. Discussion

This study evaluated the toxicity of various insecticides (including emamectin benzoate, spinosad, and chlorpyrifos) on second- and fourth-instar larvae of both laboratory and field strains of *S. littoralis*. Emamectin benzoate was found to be the most toxic insecticide for both instars, while spinosad showed the lowest toxicity. Similarly, Korrat et al. [32] found that emamectin benzoate was the most effective insecticide compared to chlorfluazuron, profenfos, and spinosad, with second-instar larvae being more susceptible than fourth-instar larvae, as indicated by the calculated LC_50_ and LC_90_ values. Deepti Pande and Srivastava [33] noted that LC_50_ values for insecticides on fourth-instar larvae were higher after 24 h compared to 48 h when compared to IGRs. Thus, emamectin benzoate and spinosad are considered standard chemicals for calculating the toxicity index and potency levels. El-Aw [34] reported that emamectin benzoate was the most toxic insecticide, followed by profenfos and spinosad. Furthermore, emamectin benzoate was found to be the most toxic insecticide against *S. littoralis*, with lower LC_50_ values when compared to Dipel and Endosulfan [35]. Haga et al. [36] observed that chlorfluazuron was highly toxic to insects due to its slow metabolism in the insect body. Chlorfluazuron demonstrated higher toxicity to *S. littoralis* compared to chlorpyrifos and profenfos after 72 h of exposure to second-instar larvae of *S. littoralis* [37]. After 72 h, the LC_50_ of spinosad was 22.179 ppm. Cypermethrin exhibited a stronger toxic effect on *S. littoralis* than chlorpyrifos. These findings align with the results of [38] which reported that emamectin benzoate was the most effective compound against fourth-instar larvae of *S. littoralis*. Our findings showed that the insecticide treatments increased larval mortality, with lufenuron and spinosad causing the highest mortality, while emamectin benzoate had the least impact. All treatments reduced larval duration, pupation, adult emergence, fecundity, and hatchability, with lufenuron causing the greatest decrease in egg production. Mohamed et al. [39] noted a negative relationship between concentration and pupation or moth emergence rates. Kandil et al. [40] found that chlorfluazuron reduced the average number of eggs deposited per female. Larval mortality was likely due to disturbances in the larval bioassay following the ingestion of leaves treated with the various insecticides [4]. Additionally, some larvae that survived after the protracted larval stage failed to reach the next advanced instar [41,42]. Our results showed that all tested insecticides reduced the weight of pupae developed from the insecticide-treated second-instar larvae compared with the control. It is assumed that the reason for the loss in weight of insecticide-treated larvae was that they were less able to convert ingested and digested food into body substances [22]. Our study indicated that all the tested combinations increased the mortality percentage to 80% in emamectin benzoate + chlorpyrifos, 100% in emamectin benzoate + cypermethrin, and 87.5% in emamectin benzoate+ lufenuron. All combinations caused a significant decrease in pupation, adult emergence, fecundity, and hatchability of the tested second-instar larvae of *S. littoralis* compared to the control. Moustafa and El-Attal [43] observed that equitoxic binary mixtures of chlorpyrifos/triflumuron (EC_25_ + EC_25_) resulted in a significant synergistic effect, as indicated by the percent inhibition of adult emergence. Similar findings were reported by El-Zahi [44], who noted that the latent effects of emamectin benzoate on the fourth-instar larvae of *S. littoralis* led to considerable reductions in pupal duration, pupal weight, pupation, and adult emergence percentages. Bio-pesticides derived from various sources are known to be effective against a wide range of arthropod pests [45]. In addition, our findings align with those of Abo El-Ghar et al. [46], who studied the effects of *Bacillus thuringiensis* and Abamectin on *S. littoralis*. Their results showed a significant reduction in pupation (36%) following abamectin treatment, along with a substantial decrease in moth fecundity (87.4%). Our study investigated the combined effects of emamectin benzoate at LC_25_ and its mixtures with chlorpyrifos, spinosad, cypermethrin, and lufenuron at LC_50_ on the second- and fourth-instar larvae of *S. littoralis*. Binary mixtures of emamectin benzoate and these insecticides were tested to determine whether their effects were antagonistic, synergistic, or additive. Our study found additive effects between emamectin benzoate and chlorpyrifos (LC_25_), lufenuron (LC_25_ and LC_50_), and cypermethrin (LC_25_) in both second- and fourth-instar *S. littoralis* larvae. Potentiation was observed with cypermethrin (LC_50_) and chlorpyrifos (LC_50_), while antagonistic effects occurred with spinosad (LC_25_ and LC_50_). The antagonistic effect between emamectin benzoate and spinosad may result from several factors, including their similar modes of action on the insect nervous system, which could lead to interference in their individual mechanisms of toxicity [47]. Additionally, *S. littoralis* may have developed partial resistance to one or both insecticides, reducing the overall efficacy of the combination. Increased detoxification via enzymes or altered insect behavior (such as reduced feeding or avoidance) could also contribute to the antagonism, as could potential effects on the penetration or uptake of the insecticides [19]. Together, these factors may prevent the two insecticides from working synergistically and instead result in a reduced overall toxicity. Korrat, Abdelmonem, Helalia, and Khalifa [32] demonstrated that combining profenfos with either emamectin benzoate or spinosad produced additive effects, while the combination of profenfos with chlorfluazuron resulted in a synergistic effect. Likewise, Radwan et al. [48] found that when spinosad and abamectin were combined with profenfos at various mixing ratios, they primarily exhibited an antagonistic effect, as determined by co-toxicity factors calculated from LC_50_ values at 24 and/or 72 h. The phenomenon of antagonism may be explained as the interference of two insecticides by severe inhibition or slowing; in both cases, the opportunity for further detoxification which generally took place after activation could be expected in this respect. The phenomenon of potentiation could be attributed to the inhibition of detoxification mechanisms by one of the two toxicants in the mixture, thus causing the high titer of the other component to react with the specific target. Each behavior depends greatly on the permeability of each toxicant via the insect integument as well as the second barrier surrounding the target (e.g., nerve sheath), which in turn depends greatly on the polarity, stability (rate of degradation), partitioning, and storage of the test compounds. Indeed, this explanation could be noticed with compounds which have similar modes of action and with those possessing an independent mode of action. Ghoneim et al. [49] reported that when chlorpyrifos was combined with chlorfluazuron, it produced an additive effect against the fourth-instar larvae of *S. littoralis*, as determined by the co-toxicity factor. El-Sheikh [47] observed that the combination of emamectin benzoate with either lufenuron or spinosad resulted in either additive or antagonistic effects, indicating that single applications of these insecticides were more effective than their combinations. Abou-Taleb et al. [50] found that the toxicity of emamectin benzoate against various larval instars of both laboratory and field strains of *S. littoralis* increased with higher concentrations and longer exposure times, but decreased with older instars. Prasad et al. [51] identified emamectin benzoate as the most toxic insecticide against *S. littoralis*. In an experiment for testing chemical and bio-insecticides on tomato, Roby and Hussein [52] showed that emamectin benzoate exhibited the highest toxic effect against both *S. littoralis* and *Tuta absoluta,* with a low effect on the beneficial insects found on tomato. They stated that the differences among the chemical and bio-insecticides tested can be attributed to the frequency of sprays and different mode of actions. Therefore, they considered that emamectin benzoate can play a good role in tomato-integrated control strategies. Chemical insecticide application continues to be the primary method for managing *S. littoralis*, with key insecticide groups used to control the pests and protect the economic crops [53]. This study suggests that enzymes such as esterase and acetylcholine esterase may play a significant role in the detoxification of synthetic pyrethroids and organophosphates in *S. litura*. Metabolic enzymes, including mixed-function oxidases, carboxylesterases, and glutathione S-transferase, are crucial in the development of insecticide resistance. Acetylcholinesterase (AChE) plays a vital role in neurotransmission by hydrolyzing acetylcholine (ACh) in cholinergic synapses, and it serves as the target site for several neurotoxic insecticides [54]. Acetylcholinesterase (AChE) is the primary target for inhibition by organophosphate and carbamate insecticides. When AChE activity is blocked, acetylcholine accumulates, leading to repeated neuronal firing and, ultimately, the death of the insect [55]. In laboratory strains, AChE was the only enzyme to show increased activity in response to emamectin benzoate. The observed decrease in AChE activity in the treated larvae suggests that the insecticides have an inhibitory effect when compared to the control. Salgado et al. [56] indicated that the overproduction of acetylcholine may occur because spinosad interacts with the nicotinic acetylcholine receptor (nAChR) and acetylcholine simultaneously, while also acting on a novel site distinct from the acetylcholine binding site. They proposed the existence of two specific binding sites on nAChR for spinetoram and acetylcholine. The increased AChE activity observed in response to organophosphates and pyrethroids suggests elevated expression levels of AChE, with target site insensitivity being a dominant mechanism contributing to resistance in lepidopteran pests [57]. Glutathione S-transferase (GST) is a diverse enzyme family involved in detoxification processes in insects [58,59]. In our study, an increase in GST activity was observed in the field strain compared to the laboratory strain, with GST being the only enzyme showing elevated activity in response to chlorpyrifos in both strains. Carboxylesterases showed the highest significant activity in response to spinosad and chlorpyrifos. These findings align with Zhang et al. [60], who found that the overexpression of GST and carboxylesterases (CarE) is associated with resistance to diamide insecticides. Furthermore, a positive correlation was noted between spinosad resistance and the increased activity of GST and carboxylesterases [61]. Carboxylesterases and GST were key factors in the development of resistance to indoxacarb in *S. exigua* and *P. xylostella* [62]. Esterases and GST have been linked to metabolic resistance in various insect pests [63]. Our study supports these findings, showing that GST and esterase activities were higher in the field strain than in the laboratory strain, consistent with Yu et al. [64], who found higher detoxification enzyme activities in the field strain of *S. frugiperda*. Esterases, crucial detoxification enzymes, hydrolyze ester bonds in synthetic chemicals [65] and contribute to pyrethroid resistance in noctuids, especially *H. armigera* [66]. Notably, IGRs caused significant changes in alpha and beta esterase levels in *S. littoralis* [67]. Spinetoram slightly increased alpha esterase activity compared to the control, while emamectin benzoate led to a significant increase in beta esterase [54]. Siqueira et al. [68] highlighted the role of esterase in abamectin resistance in the Brazilian *T. absoluta* population, confirming the insecticide’s detoxification. Zibaee [69] showed that the exposure of *T. absoluta* fourth-instar larvae to chlorpyrifos elevated esterase activity after 24 and 48 h, which was consistent with the toxicity ratios of the mixtures. In our study, treatment with the mixtures of emamectin benzoate with either lufenuron or spinosad increased alpha esterase activity, suggesting that this enzyme plays a critical role in detoxifying these mixtures, contributing to their high toxicity to *S. littoralis*. We also observed that these mixtures suppressed the activity of other detoxifying enzymes, such as GST. Badawy et al. [70] reported that GST and carboxylesterase were involved in the detoxification of low doses of pymetrozine and acetamiprid in *A. mellifera*. Protein levels are essential in insect growth, development, and metabolism [71]. Our results show a decrease in total protein content in the fourth-instar larvae of *S. littoralis* treated with lufenuron, an IGR, which aligns with findings by Awadalla et al. [72]. This reduction may be due to the inhibition of DNA and RNA synthesis, as seen in other studies [73]. In our study, the decrease in total protein content was also observed in larvae treated with mixtures of emamectin benzoate and either lufenuron or cypermethrin, indicating that these mixtures have high toxicity. Generally, the increased activity of detoxifying enzymes, such as esterases, is a common resistance mechanism in insects, particularly for organophosphates and carbamates [74]. In other words, our findings present novel insights by evaluating the field efficacy and enzymatic responses of *S. littoralis* to specific insecticide combinations with emamectin benzoate, an approach that integrates biochemical and field-based assessments to inform more effective and economical pest management strategies.

## 5. Conclusions

This study demonstrates that emamectin benzoate is the most effective insecticide against *Spodoptera littoralis* (cotton leafworm) in cotton crops, with an average effectiveness greater than 80%. Lufenuron also exhibited high efficacy, while spinosad showed the least effectiveness. The findings indicate that while mixtures of insecticides, such as emamectin benzoate + lufenuron, can enhance control effectiveness, they also require higher quantities of insecticides and result in lower overall control compared to individual applications. Therefore, the use of insecticides individually, particularly emamectin benzoate and lufenuron, is more cost-effective and environmentally sustainable for managing cotton leafworm infestations. Additionally, the study highlighted the influence of different insecticides on enzymatic activities in larvae, with significant increases observed in α-esterase, β-esterase, carboxylesterase, acetylcholinesterase, and glutathione S-transferase activities, indicating the potential biochemical impacts of insecticide treatments. These results support the optimization of insecticide application strategies, emphasizing the importance of the selective and individual use of insecticides to maximize pest control while minimizing environmental and economic costs.

## Figures and Tables

**Figure 1 insects-16-00821-f001:**
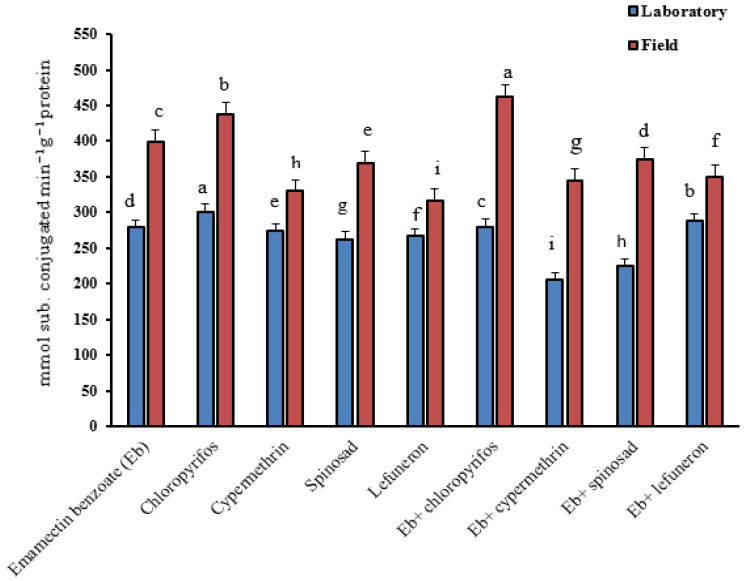
The glutathione S-transferase activity in *S. littoralis* larvae treated with insecticides alone and their combinations against fourth-instar larvae of the field strain and laboratory strain of *S. littoralis*. Values are shown as the mean ± SE of three replicates. Numbers followed by different letters within the same column indicate that the data are statistically different at *p* < 0.05.

**Figure 2 insects-16-00821-f002:**
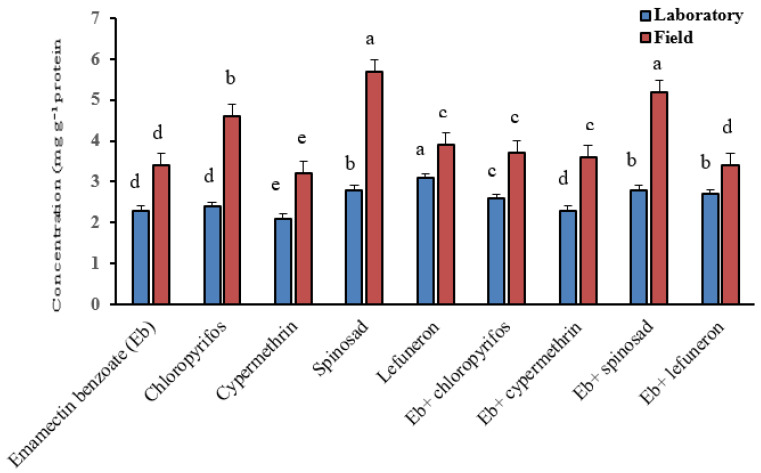
Total protein concentrations in *S. littoralis* larvae treated with insecticides alone and their combinations against fourth-instar larvae of the field strain and the laboratory strain of *S. littoralis*. Values are shown as the mean ± SE of three replicates. Numbers followed by different letters within the same column indicate that the data are statistically different at *p* < 0.05.

**Table 1 insects-16-00821-t001:** Toxicity of the tested insecticides against second and fourth larval instars on laboratory and field strains of *S. littoralis* under laboratory conditions.

Insecticides	Strain	LC_50_ (μg/mL) (95%CL)	LC_90_ (μg/mL) (95%CL)	Slope	Toxicity Index **		Relative Potency ***		RR *
					LC_50_	LC_90_	LC_50_	LC_90_	
2nd instar larvae									
Emamectin benzoate	Laboratory	0.01 (0.0001–0.02)	0.08 (0.07–0.35)	0.17	100	100	1240	513.75	1.00
	Field	0.01 (0.01–0.02)	0.11 (0.03–30.01)	0.46	100	100	1636	770	13.6
Lufenuron	Laboratory	1.23 (0.71–1.99)	26.04 (1.90–3.83)	0.87	0.81	0.31	10.08	1.58	6.72
	Field	3.06 (1.84–4.96)	125.68 (21.47–240.20)	0.90	0.12	0.09	1.98	0.67	4.83
Cypermethrin	Laboratory	3.81 (2.18–6.25)	47.15 (26.86–116.95)	0.93	0.26	0.17	2.17	0.87	1.32
	Field	5.02 (3.42–7.25)	93.94 (39.17–505.99)	0.97	0.2	0.23	3.26	0.90	2.00
Chlorpyrifos	Laboratory	12.40 (9.82–15.01)	41.1 (35.93–79.88)	0.96	0.08	0.19	1.00	1.00	1.32
	Field	16.36 (12.49–21.31)	84.77 (62.99–127.49)	0.95	0.06	0.13	1.88	1.00	2.06
Spinosad	Laboratory	5.72 (3.30–10.24)	125.86 (51.49–98.93)	0.88	0.17	0.06	3.25	0.33	1.45
	Field	8.27 (5.06–13.59)	185.29 (63.40–173.0)	0.84	0.12	0.06	1.98	0.46	1.47
4th instar larvae									
Emamectin benzoate	Laboratory	0.05 (0.05–0.07)	0.13 (0.11–0.21)	0.98	100	100	583.3	599	1.2
	Field	0.06 (0.05–0.07)	0.15 (0.11–0.029)	0.86	100	100	613.83	762.47	1.15
Lufenuron	Laboratory	4.82 (1.81–7.34)	31.34 (20.65–80.90)	0.99	1.03	0.15	6.06	2.49	2.92
	Field	14.08 (9.85–20.82)	133.38 (68.17–475.69)	0.86	0.34	0.11	2.61	0.86	4.26
Cypermethrin	Laboratory	8.99 (2.18–6.25)	113.36 (61.84–604.35)	0.85	0.56	0.14	3.25	0.69	1.63
	Field	14.62 (3.42–7.25)	115.47 (74.11–604.35)	0.87	1.30	0.13	2.58	0.99	0.41
Chlorpyrifos	Laboratory	29.19 (13.89–19.06)	77.90 (65.65–121.19)	0.98	0.17	0.12	1.00	1.0	1.26
	Field	36.80 (30.80–45.04)	114.37 (103.66–211.44)	0.94	0.65	0.13	1.00	1.0	0.37
Spinosad	Laboratory	25.08 (18.80–32.88)	118.06 (73.21–330.53)	0.87	0.20	0.11	1.16	0.67	1.16
	Field	29.09 (21.58–40.62)	163.32 (90.06–684)	0.99	0.24	0.09	1.0	4.56	0.21

* RR resistance ratio calculated as LC_50_ of field strain/LC_50_ of laboratory strain. ** Toxicity index calculated as (LC_50_ or LC_90_ of the highest efficient compound/LC_50_ or LC_90_ of the other compound) × 100. *** Relative potency calculated as LC_50_ or LC_90_ of the lowest efficient compound/LC_50_ or LC_90_ of the other compound.

**Table 2 insects-16-00821-t002:** Biological effect of emamectin benzoate at LC_25_ and chlorpyrifos, spinosad, cypermethrin, and lufenuron at LC_50_ and their combinations against the second larval instar of *S. littoralis* under laboratory conditions.

Treatment	%Larval Mortality	Larval Duration (Days)	Pupal Duration (Days)	Pupation%	Pupal Mean Weight (mg)	Adult Emergence (%)	Fecundity (No. Eggs Laid/Female)	Fertility (%Egg Hatched)	Incubation Period (Days)	Adult Longevity (Days)
Control	0 ± 0	19 ± 0.90 ^ab^	9.7 ± 0.70 ^b^	100 ^a^	2.71 ± 0.63 ^a^	100 ^a^	1125 ± 82.91 ^a^	98.67 ^a^	2.75 ± 0.43 ^b^	8 ± 0.70 ^ab^
Emamectin benzoate	27.5	16.79 ± 0.16 ^c^	9.7 ± 0.10 ^b^	72.5 ^b^	1.93 ± 0.15 ^b^	70 ^b^	862.5 ± 96.01 ^b^	75.36 ^c^	3.0 ± 0.71 ^b^	8 ± 0.71 ^ab^
Chlorpyrifos	50	18.6 ± 0.14 ^b^	9.6 ± 0.14 ^b^	50 ^c^	1.55 ± 0.21 ^c^	50 ^c^	637.5 ± 119.24 ^c^	70.58 ^e^	3.5 ± 0.5 ^b^	7 ± 0.70 ^c^
Emamectin + chlorpyrifos	80	17.63 ± 0.22 ^b^	10.88 ± 0.21 ^a^	20 ^g^	0.544 ± 0.06 ^c^	20 ^g^	500 ± 40.82 ^b^	48 ^f^	3.67 ± 0.47 ^a^	6 ± 82 ^b^
Cypermethrin	47.5	18.67 ± 0.13 ^b^	9.77 ± 0.23 ^b^	47.5 ^d^	0.94 ± 0.11 ^e^	47.5 ^d^	600 ± 35.35 ^c^	75 ^c^	3.25 ± 0.0.43 ^a^	7.25 ± 0.43 ^b^
Emamectin + cypermethrin	75.5	18.00	10.00 ± 0.31 ^a^	50 ^c^	0.96 ± 0.12 ^e^	50 ^c^	650.5 ± 55.0 ^c^	50 ^f^	2.75 ± 0.43 ^b^	7.0 ± 0.5 ^ab^
Spinosad	45	19.55 ± 0.05 ^a^	9.56 ± 0.13 ^b^	45 ^e^	1.02 ± 0.06 ^d^	45 ^e^	737.5 ± 96.01 ^bc^	77.28 ^b^	3.0 ± 0.70 ^b^	8.5 ± 0.5 ^a^
Emamectin + spinosad	60	17.71 ± 0.32 ^ab^	9.5 ± 0.18 ^c^	40 ^f^	1.09 ± 0.07 ^d^	40 ^b^	516 ± 62.63 ^b^	38.75 ^c^	3.3 ± 0.47 ^ab^	7.3 ± 0.47 ^ab^
Lufenuron	50	19.6 ± 0.14 ^a^	10.25 ± 0.31 ^a^	50 ^c^	1.23 ± 1.19 ^d^	42.5 ^f^	625 ± 55.90 ^c^	72 ^d^	2.75 ± 0.43 ^b^	7.5 ± 0.5 ^ab^
Emamectin + lufenuron	87.5	18.25 ± 0.43 ^ab^	9.75 ± 0.43 ^b^	12.5 ^h^	0.225 ± 0.19 ^f^	12.5 ^h^	450 ± 40.82 ^d^	42.1 ^g^	2.67 ± 0.47 ^ab^	7.0 ± 0.82 ^ab^

Values are shown as mean ± SD of three replicates. Numbers followed by different letters within the same column indicate that data are statistically different at *p* < 0.05.

**Table 3 insects-16-00821-t003:** Joint action of emamectin benzoate at the level of LC_25_ and its mixture with LC_25_ chlorpyrifos, spinosad, cypermethrin, and lufenuron at the level of LC_50_ against the second- and fourth-instar larvae of *S. littoralis*.

Combinations	% Mortality	Co-Toxicity Factor	Effect
2nd instar larvae			
Emamectin benzoate LC_25_ + chlorpyrifos LC_25_	43.33	−18.75	Additive
Emamectin benzoate LC_25_ + lufenuron LC_25_	56.67	6.26	Additive
Emamectin benzoate LC_25_ + cypermthrin LC_25_	46.67	−6.66	Additive
Emamectin benzoate LC_25_ + spinosad LC_25_	30	−40	Antagonism
Emamectin benzoate LC_25_ + chlorpyrifosLC_50_	83.33	24.99	potentiation
Emamectin benzoate LC_25_ + lufenuron LC_50_	70	5.01	Additive
Emamectin benzoate LC_25_ + cypermthrin LC_50_	80	21.06	Potentiation
Emamectin benzoate LC_25_ + spinosad LC_50_	65	−35	Antagonism
4th instar larvae			
Emamectin benzoate LC_25_ + spinosad LC_50_	33.33	−33.34	Antagonism
Emamectin benzoate LC_25_ + chlorpyrifos LC_25_	53.33	6.66	Additive
Emamectin benzoate LC_25_ + lufenuron LC_25_	50	7.13	Additive
Emamectin benzoate LC_25_ + cypermthrin LC_25_	43.33	−13.34	Additive
Emamectin benzoate LC_25_ + spinosad LC_25_	40	−25	Antagonism
Emamectin benzoate LC_25_ + chlorpyrifos LC_50_	76.67	21	Potentiation
Emamectin benzoate LC_25_ + lufenuron LC_50_	70	10.53	Additive
Emamectin benzoate LC_25_ + cypermthrin LC_50_	83.33	31.58	Potentiation

**Table 4 insects-16-00821-t004:** Field efficiency of the tested insecticides and their mixtures against *Spodoptera littoralis* on cotton during the 2023 season.

Treatments		Pre-Count	% Mortality		Residual Effect		Residual Mean of % Reduction	Annual Mean% of Reduction
			1 Day	3 Days	7 Days	10 Days		
2023								
Emamectin benzoate (EB)	No	153.33 ± 1.53	22 ± 2.00	***	19 ± 1.00	13.33 ± 1.53	89.91 ± 0.70 ^c^	88.57 ± 0.85 ^d^
	%		85.90 ± 1.15	***	88.90 ± 0.54	91.80 ± 0.90		
Chlorpyrifos	No	181.33 ± 1.53	30.67 ± 1.53	***	28 ± 2.00	24.67 ± 1.53	86.12 ± 0.86 ^g^	85.21 ± 0.81 ^e^
	%		83.37 ± 0.37	***	85.08 ± 0.96	87.17 ± 0.76		
Cypermethrin	No	158.00 ± 2.00	28 ± 2.00	***	24 ± 1.00	21.67 ± 1.53	86.19 ± 0.64 ^g^	85.00 ± 0.87 ^e^
	%		82.59 ± 1.03	***	85.32 ± 0.46	87.07 ± 0.82		
Spinosad	No	158.00 ± 2.00	29 ± 2.00	***	25.67 ± 1.53	22.67 ± 1.53	85.39 ± 0.80 ^g^	84.24 ± 0.87 ^e^
	%		81.96 ± 1.02	***	84.30 ± 0.81	86.47 ± 0.81		
Lufenuron	No	163.33 ± 2.52	**	23 ± 2.00	22 ± 1.00	19 ± 1.00	88.00 ± 0.43 ^f^	87.44 ± 0.62 ^d^
	%		**	86.30 ± 0.99	86.98 ± 0.43	89.03 ± 0.47		
EB + chlorpyrifos	No	166.00 ± 1.00	10 ± 1.00	8 ± 1.00	6 ± 1.00	4.76 ± 0.58	96.84 ± 0.28 ^a^	95.98 ± 0.47 ^a^
	%		94.08 ± 0.56	59.29 ± 0.56	95.49 ± 0.53	97.18 ± 0.03		
EB + cypermethrin	No	155.67 ± 1.53	20.33 ± 1.53	17.33 ± 2.08	14 ± 1.00	11.67 ± 1.53	92.12 ± 0.71 ^d^	90.47 ± 0.36 ^c^
	%		87.16 ± 1.83	89.16 ± 1.21	91.31 ± 0.53	92.93 ± 0.89		
EB + spinosad	No	169.00 ± 1.00	18.33 ± 1.53	16 ± 1.0	13 ± 1.00	9 ± 1.00	93.75 ± 0.54 ^c^	92.28 ± 0.63 ^b^
	%		89.34 ± 0.81	90.78 ± 0.52	92.56 ± 0.53	94.94 ± 0.56		
EB + lufenuron	No	151.00 ± 2.00	15.33 ± 1.53	11 ± 1.08	9 ± 1.00	6 ± 1.00	92.25 ± 0.58 ^b^	93.51 ± 0.67 ^b^
	%		90.02 ± 0.85	89.25 ± 0.47	94.25 ± 0.57	96.25 ± 0.60		
Control	No	173.67 ± 1.53	176.33 ± 1.53	178.33 ± 1.52	180.33 ± 1.52	183.67 ± 1.52		
LSD 0.05							1.09	1.24
2024								
Emamectin benzoate (EB)	No	96.67 ± 6.03	13.33 ± 1.53	***	10.67 ± 2.08	8.67 ± 0.58	91.08 ± 1.51 ^b^	89.57 ± 1.65 ^c^
	%		86.57 ± 1.91	***	89.94 ± 3.23	92.22 ± 0.97		
Chlorpyrifos	No	100.67 ± 2.52	20 ± 1.00	***	15 ± 1.00	12.67 ± 0.64	87.82 ± 0.82 ^c^	85.45 ± 0.46 ^d^
	%		80.73 ± 0.63	***	86.46 ± 0.64	89.14 ± 0.34		
Cypermethrin	No	95.33 ± 3.51	15 ± 1.00	***	14.00 ± 1.00	13.33 ± 1.51	87.31 ± 0.62 ^c^	86.46 ± 0.50 ^d^
	%		84.75 ± 0.43	***	86.69 ± 0.54	87.93 ± 0.76		
Spinosad	No	106 ± 2.65	20 ± 1.00	***	17.00 ± 1.00	14.67 ± 0.58	86.75 ± 0.53 ^c^	85.07 ± 0.26 ^d^
	%		81.70 ± 0.58	***	85.46 ± 0.67	88.04 ± 0.76		
Lufenuron	No	98.00 ± 2.00	**	15.33 ± 0.58	12.33 ± 0.58	10.33 ± 0.58	89.72 ± 0.74 ^b^	88.27 ± 0.77 ^c^
	%		**	85.33 ± 0.89	88.58 ± 0.78	90.89 ± 0.64		
EB + chlorpyrifos	No	102.67 ± 2.52	9 ± 1.00	7.67 ± 1.53	6.67 ± 1.53	6.00 ± 1.00	94.55 ± 0.99 ^a^	93.54 ± 0.98 ^a^
	%		91.51 ± 0.69	92.49 ± 0.61	94.13 ± 1.42	94.97 ± 0.75		
EB + cypermethrin	No	97.67 ± 0.58	13 ± 2.00	10.33 ± 1.53	8.00 ± 1.00	7.67 ± 0.58	92.90 ± 0.70 ^a^	91.19 ± 1.11 ^b^
	%		87.10 ± 1.85	89.90 ± 0.85	92.57 ± 0.93	93.22 ± 0.50		
EB + spinosad	No	95.33 ± 2.08	14.67 ± 0.58	13 ± 1.00	10.33 ± 0.58	8.33 ± 0.58	90.80 ± 0.87 ^b^	89.17 ± 0.07 ^c^
	%		84.88 ± 0.32	84.88 ± 1.20	90.18 ± 0.37	91.42 ± 1.44		
EB + lufenuron	No	101 ± 2.65	9.67 ± 0.58	10.67 ± 1.15	8.00 ± 1.00	6.33 ± 1.15	93.70 ± 0.85 ^a^	92.76 ± 0.74 ^a^
	%		90.72 ± 0.54	90.08 ± 1.21	92.74 ± 0.90	94.60 ± 0.88		
Control	No	95 ± 1.00	98.33 ± 1.53	178.33 ± 1.52	104.33 ± 0.58	183.67 ± 1.52		
LSD 0.05							1.47	1.46

Values are shown as mean ± SD of three replicates. Numbers followed by different letters within the same column indicate that the data are statistically different at *p* < 0.05. ** % Mortality of insecticide recorded after one day. ******* % Mortality of IGR recorded after 3 days.

**Table 5 insects-16-00821-t005:** General esterase, carboxylesterase, and acetylcholine esterase activity in fourth-instar larvae of *Spodoptera littoralis* tested with LC_50_ of insecticides and their mixtures.

Insecticides	Strains	Esterases Activity (µg α/β naphthol/min/g protein)		Carboxylesterase Activity (µg product/min/g protein)		AChE Activity (µg product/min/g protein)
		α-NA	β-NA	α-NA	β-NA	
Emamectin benzoate (EB)	Laboratory	153.93 ± 2.3 ^d^	129.94 ± 1.3 ^b^	543.5 ± 2.8 ^b^	27.3 ± 1.3 ^g^	914.2 ± 4.1 ^f^
	Field	471.99 ± 3.1 ^d^	190.14 ± 1.5 ^g^	758.8 ± 3.1 ^c^	49.4 ± 1.4 ^h^	1315.9 ± 2.7 ^b^
Chlorpyrifos	Laboratory	154.97 ± 1.1 ^d^	135.16 ± 1.81 ^a^	230.7 ± 1.4 ^f^	36.4 ± 0.3 ^d^	1076.0 ± 3.8 ^a^
	Field	614.15 ± 3.4 ^b^	239.80 ± 1.7 ^c^	689.3 ± 1.3 ^e^	76.3 ± 0.3 ^d^	1346.8 ± 1.4 ^b^
Cypermethrin	Laboratory	157.05 ± 2.2 ^c^	129.24 ± 2.3 ^b^	151.6 ± 2.3 ^h^	45.9 ± 0.7 ^b^	1073.9 ± 2.8 ^b^
	Field	795.97 ± 1.2 ^c^	130.97 ± 2.3 ^h^	255.4 ± 1.8 ^i^	89.6 ± 0.6 ^c^	1047.9 ± 3.3 ^g^
Spinosad	Laboratory	156.09 ± 1.7 ^c^	126.28 ± 1.5 ^c^	590.0 ± 2.5 ^a^	22.3 ± 1.7 ^h^	1017.8 ± 2.6 ^e^
	Field	506.55 ± 3.4 ^e^	247.58 ± 1.3 ^b^	884.7 ± 3.4 ^a^	56.4 ± 1.3 ^f^	1428.7 ± 3.6 ^a^
Lufenuron	Laboratory	155.32 ± 3.6 ^c^	119.84 ± 1.8 ^e^	267.6 ± 1.2 ^e^	32.5 ± 1.6 ^e^	863.0 ± 4.7 ^g^
	Field	371.79 ± 4.1 ^b^	232.71 ± 1.4 ^f^	839.7 ± 1.3 ^b^	98.6 ± 1.4 ^b^	1127.5 ± 3.9 ^f^
EB + chlorpyrifos	Laboratory	143.67 ± 1.9 ^f^	118.60 ± 1.6 ^e^	298.8 ± 1.5 ^d^	57.4 ± 0.7 ^a^	1037.8 ± 3.8 ^c^
	Field	485.49 ± 2.5 ^a^	337.85 ± 1.7 ^d^	319.5 ± 2.4 ^h^	106.7 ± 0.8 ^a^	1189.1 ± 1.9 ^e^
EB + cypermethrin	Laboratory	144.84 ± 1.8 ^e^	118.65 ± 1.9 ^e^	129.6 ± 2.4 ^i^	29.3 ± 1.2 ^f^	1007.8 ± 1.7 ^c^
	Field	379.41 ± 1.4 ^b^	233.44 ± 1.7 ^e^	596.7 ± 3.1 ^f^	72.7 ± 1.4 ^e^	1298.9 ± 2.3 ^c^
EB + spinosad	Laboratory	161.69 ± 1.6 ^b^	122.44 ± 1.6 ^d^	426.3 ± 3.2 ^c^	41.3 ± 1.3 ^c^	784.1 ± 3.2 ^i^
	Field	282.31 ± 2.5 ^c^	189.71 ± 1.1 ^g^	698.1 ± 2.6 ^d^	69.1 ± 1.1 ^g^	1038.4 ± 1.6 ^h^
EB + lufenuron	Laboratory	163.65 ± 2.6 ^a^	127.32 ± 1.3 ^c^	192.9 ± 3.1 ^g^	23.6 ± 0.6 ^h^	855.1 ± 1.6 ^h^
	Field	215.81 ± 3.4 ^d^	253.44 ± 1.4 ^a^	321.7 ± 2.4 ^g^	55.7 ± 0.9 ^f^	1262.3 ± 4.3 ^d^

Values are shown as mean± SD of three replicates. Numbers followed by different letters within the same column indicate that data are statistically different at *p* < 0.05.

## Data Availability

The original contributions presented in this study are included in the article. Further inquiries can be directed to the corresponding authors.

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
