# Peer review of "Comparative Toxicological Effects of Insecticides and Their Mixtures on Spodoptera littoralis (Lepidoptera: Noctuidae)"

_insects, 2025, doi:10.3390/insects16080821_

Round 1
Reviewer 1 Report
Comments and Suggestions for Authors
Spodoptera littoralis is a major pest affecting cotton crop yield and quality. This study has practical significance and potential value for developing more effective management strategies for Spodoptera littoralis pests. In particular, the results show that the use of a single insecticide is more cost-effective, which is an important reference for agricultural producers with limited resources. This study fills the gap in existing knowledge, especially in evaluating the comparative effects of different insecticide combinations on Spodoptera littoralis management. However, several aspects require further clarification and improvement before publication consideration.
Major comments:
- The article may not provide sufficient detail in the methodology section and needs to supplement the specific methods of experimental design, concentrations of insecticides, and data analysis to enhance the credibility and replicability of the study.
- Experimental control conditions: The article mentioned that distilled water was used as a control, but it did not elaborate on whether the setting of the control group was consistent in all experiments and whether the interfering factors that might affect the interpretation of the results were fully considered. It is recommended that the authors clarify and explain in detail the setting of the control group to ensure the accuracy of the results.
- The description of laboratory and field conditions meets the needs of the research objectives. However, environmental factors such as climate and soil characteristics that may affect the results of field experiments need to be further described to evaluate whether these variables may cause bias in the results. It is recommended that the authors supplement this information to ensure the rigor of the study.
- In the discussion of the experimental results, it is necessary to combine real-world cases and existing literature to specifically point out the novelty of this study, as well as its significance and application potential in cotton crop pest management practice.
Other comments:
- The term "alfa-esterase" should be "alpha-esterase" in the abstract to maintain consistency throughout the document.
- L102-103. "Chemicals that used in determination of acetycholine esterase purchased from sigma Aldrich chemical Ltd., Cairo, Egypt." should be "Chemicals that were used in determination of acetylcholine esterase were purchased from Sigma-Aldrich Chemical Ltd., Cairo, Egypt."
- The phrase "Insecticides applications" should be corrected to "Insecticide applications".
- "at the rate of 200 liter per fed." should be "at the rate of 200 liters per feddan."
- In the main content, "instar d higher level" should be corrected to "instar had a higher level" for clarity.
- L332 and L337. In the Figure/Table Captions, consistent formatting should be employed (e.g., "Table (1):" should be "Table 1." to match other table notations).
- References [6], [7], [14], [17], [25], [26], [34] etc. are missing citation details. Please ensure all references are complete and properly formatted.
Author Response
Response to Reviewer 1
Thank you so much for your insightful feedback; we truly appreciate the time and effort you’ve dedicated to helping us. We are especially grateful, as you not only provide comments but also guide us toward effective solutions. We have made all the suggested revisions and are hopeful that these changes align with your expectations.
First: The manuscript has been edited by MDPI services (please see the certificate)
- Comment: The article may not provide sufficient detail in the methodology section and needs to supplement the specific methods of experimental design, concentrations of insecticides, and data analysis to enhance the credibility and replicability of the study.
Response: This correction has been made.
- Comment: control conditions: The article mentioned that distilled water was used as a control, but it did not elaborate on whether the setting of the control group was consistent in all experiments and whether the interfering factors that might affect the interpretation of the results were fully considered. It is recommended that the authors clarify and explain in detail the setting of the control group to ensure the accuracy of the results.
Response: This has been done according to your valuable comment (135-156).
- Comment: The description of laboratory and field conditions meets the needs of the research objectives. However, environmental factors such as climate and soil characteristics that may affect the results of field experiments need to be further described to evaluate whether these variables may cause bias in the results. It is recommended that the authors supplement this information to ensure the rigor of the study.
Response: This correction has been made (Lines 124-126; 179-181; 184-186).
- Comment: In the discussion of the experimental results, it is necessary to combine real-world cases and existing literature to specifically point out the novelty of this study, as well as its significance and application potential in cotton crop pest management practice.
Response: Done.
- Comment: The term "alfa-esterase" should be "alpha-esterase" in the abstract to maintain consistency throughout the document.
Response: Modified in line 45.
- Comment: L102-103. "Chemicals that used in determination of acetycholine esterase purchased from sigma Aldrich chemical Ltd., Cairo, Egypt." should be "Chemicals that were used in determination of acetylcholine esterase were purchased from Sigma-Aldrich Chemical Ltd., Cairo, Egypt."
Response: Modified in line 119.
- Comment: The phrase "Insecticides applications" should be corrected to "Insecticide applications"
Response: Corrected in line 194.
- Comment: "at the rate of 200 liter per fed." should be "at the rate of 200 liters per feddan."
Response: Done in line 196.
- Comment: In the main content, "instar d higher level" should be corrected to "instar had a higher level" for clarity.
Response: Corrected in line 271.
- Comment: L332 and L337. In the Figure/Table Captions, consistent formatting should be employed (e.g., "Table (1):" should be "Table 1." to match other table notations).
Response: Done.
- Comment: References [6], [7], [14], [17], [25], [26], [34] etc. are missing citation details. Please ensure all references are complete and properly formatted.
Response: Done.

Reviewer 2 Report
Comments and Suggestions for Authors
Testing insecticides against insects in laboratory and field is a necessary step in evaluation of effectiveness of pest control measures. In vivo and in vitro studies exploiting biochemical and molecular approaches are inevitable for elucidation of mechanisms of insecticide action. I just can’t understand what “in vivo” and “in vitro” imply, as these terms cannot be found beyond the title of the manuscript.
This is logical to expect different interactions of insecticides of different chemical groups (classes), and there are numerous examples of both antagonism & synergy. The fact that increased activity of counteracting insecticides could not be achieved among the combinations tested is the main drawback of the study. More robust explanations OR new experiments showing more promising combinations are expected.
Reading legends to figures becomes a challenge as one can find labels which are not clear (laboratory, field) or too complicated (axis Y) for perception. Meanwhile, key parameters remain unexplained there. For example, strains of what, and what kind of strains, are considered in Figures 1 & 2?
To decide, whether the research is actual at the modern level of pest control using synthetic insecticides, this is essential to understand the range of active compounds tested in terms of their mode of action, distribution range, application history, pest resistance levels achieved so far, etc. Introduction explicitly covers some of these aspects while others remain undisclosed. The authors may think on a more profound Introduction & Discussion to explain the background of the insecticides utilized and substantiate their use in the current study. Conversely, there’s no need to repeat Results in Discussion so it can be modified without lengthening.
This is obvious and well-known that the older instars are less sensitive to pesticides than the younger ones, no need focusing on that in Discussion.
This is also logical to expect higher values of LC50 in first days as compared to the last ones (if I get it right and this simply means that effect of insecticides is more profound over time) and no need to highlight it in Discussion.
In Discussion, what does it mean “similar trends for fold numbers”?
It is quite clear that obtained results corroborate with the previous studies, but the novelty of the current paper remains undisclosed.
The most important drawback is that the authors only tested insecticide combinations which showed neutral or negative effect of their combination. It would be more interesting for the journal’s readers to learn which combinations are more effective than the single compound.
Author Response
Response to Reviewer 2
We sincerely thank you for your thoughtful and constructive feedback on our manuscript. Your detailed comments and suggestions have been invaluable in helping us improve the clarity, quality, and overall impact of our work. We truly appreciate the time and effort you invested in reviewing our submission, and your insights have significantly contributed to strengthening our study. Thank you again for your guidance and support throughout the revision process.
First: The manuscript has been edited by MDPI services (please see the certificate)
- Comment: Testing insecticides against insects in laboratory and field is a necessary step in evaluation of effectiveness of pest control measures. In vivo and in vitro studies exploiting biochemical and molecular approaches are inevitable for elucidation of mechanisms of insecticide action. I just can’t understand what “in vivo” and “in vitro” imply, as these terms cannot be found beyond the title of the manuscript.
Response: Our manuscript focused on the evaluation of insecticide toxicity against Spodoptera littoralis under field and laboratory conditions. We use “In vivo” to mean a field experiment and “In vitro” for a laboratory experiment. However, we have removed “in vivo” and “in vitro” from the title.
- Comment:This is logical to expect different interactions of insecticides of different chemical groups (classes), and there are numerous examples of both antagonism & synergy. The fact that increased activity of counteracting insecticides could not be achieved among the combinations tested is the main drawback of the study. More robust explanations OR new experiments showing more promising combinations are expected.
Response: Thank you for your comment. An actual increase in counteracting insecticide toxicity has been observed among the combinations tested. Table 3 shows that there is an additive effect between emamectin benzoate with chlorpyrifos (LC25), lufenuron (LC25 and LC50), and cypermethrin (LC25) in the 2nd and 4th instar larvae of S. littoralis. Notably, potentiation was observed in the interactions between emamectin benzoate with cypermethrin (LC50) and chlorpyrifos (LC50) in the 2nd and 4th instar larvae of S. littoralis. However, antagonistic effects were recorded in the combination of emamectin benzoate with spinosad (LC25 and LC50) in the 2nd and 4th instar larvae of S. littoralis. These findings highlight the complexity of insecticide interactions, with varying outcomes depending on the chemical combinations and larval developmental stages.
The antagonistic effect between emamectin benzoate and spinosad may result from several factors, including their similar modes of action on the insect nervous system, which could lead to interference in their mechanisms of toxicity. Additionally, Spodoptera littoralis may have developed partial resistance to one or both insecticides, reducing the overall efficacy of the combination. Increased detoxification via enzymes or altered insect behavior (such as reduced feeding or avoidance) could also contribute to the antagonism, as could potential effects on the penetration or uptake of the insecticides. Together, these factors may prevent the two insecticides from working synergistically and instead result in a reduced overall toxicity. This point has been clarified in the Discussion section (Lines 527 to 539).
- Comment: Reading legends to figures becomes a challenge as one can find labels which are not clear (laboratory, field) or too complicated (axis Y) for perception. Meanwhile, key parameters remain unexplained there. For example, strains of what, and what kind of strains, are considered in Figures 1 & 2?
Response: The captions of the figures have been modified.
- Comment: To decide, whether the research is actual at the modern level of pest control using synthetic insecticides, this is essential to understand the range of active compounds tested in terms of their mode of action, distribution range, application history, pest resistance levels achieved so far, etc. Introduction explicitly covers some of these aspects while others remain undisclosed. The authors may think on a more profound Introduction & Discussion to explain the background of the insecticides utilized and substantiate their use in the current study. Conversely, there’s no need to repeat Results in Discussion so it can be modified without lengthening.
Response: Done in lines 70-85. We have also removed the repeated results and rewritten them according to your valuable comments.
- Comment: This is obvious and well-known that the older instars are less sensitive to pesticides than the younger ones, no need focusing on that in Discussion.
Response: Removed.
- Comment: This is also logical to expect higher values of LC50 in first days as compared to the last ones (if I get it right and this simply means that effect of insecticides is more profound over time) and no need to highlight it in Discussion.
Response: Done accordingly.
- Comment: In Discussion, what does it mean “similar trends for fold numbers”?
Response: We are sorry for this mistake and have modified it.
- Comment: It is quite clear that obtained results corroborate with the previous studies, but the novelty of the current paper remains undisclosed.
Response: Thank you for your valuable comment. A description of the novelty of our manuscript has been added in the Abstract (Lines 29-31) and the Discussion (Lines 618-622).
- Comment: The most important drawback is that the authors only tested insecticide combinations which showed neutral or negative effect of their combination. It would be more interesting for the journal’s readers to learn which combinations are more effective than the single compound.
Response: Thank you for your comment. An actual increase in counteracting insecticide toxicity has been observed among the combinations tested. Table 3 shows that there is an additive effect between emamectin benzoate with chlorpyrifos (LC25), lufenuron (LC25 and LC50), and cypermethrin (LC25) in the 2nd and 4th instar larvae of S. littoralis. Notably, potentiation was observed in the interactions between emamectin benzoate with cypermethrin (LC50) and chlorpyrifos (LC50) in the 2nd and 4th instar larvae of S. littoralis. However, antagonistic effects were recorded in the combination of emamectin benzoate with spinosad (LC25 and LC50) in the 2nd and 4th instar larvae of S. littoralis. These findings highlight the complexity of insecticide interactions, with varying outcomes depending on the chemical combinations and larval developmental stages.
The antagonistic effect between emamectin benzoate and spinosad may result from several factors, including their similar modes of action on the insect nervous system, which could lead to interference in their mechanisms of toxicity. Additionally, Spodoptera littoralis may have developed partial resistance to one or both insecticides, reducing the overall efficacy of the combination. Increased detoxification via enzymes or altered insect behavior (such as reduced feeding or avoidance) could also contribute to the antagonism, as could potential effects on the penetration or uptake of the insecticides. Together, these factors may prevent the two insecticides from working synergistically and instead result in a reduced overall toxicity. This point has been clarified in the Discussion section (Lines 527 to 539).

Round 2
Reviewer 1 Report
Comments and Suggestions for Authors
The authors have adequately and reasonably addressed my comments
Author Response
The authors have adequately and reasonably addressed my comments
Response: Thank you.
Reviewer 2 Report
Comments and Suggestions for Authors
The paper is significantly improved and the explanations given in response to the reviewer’s comments are mainly satisfactory. This makes the manuscript considerable for publication, requiring more attentive reading of the text. Newly found gaps are given below.
First of all, why control is absent from all data sheets and graphs?
L2-3: Toxicology …. on Spodoptera – unclear wording
L32: aim .. investigates – a wrong choice of subject vs predicate
L38: it is not clear what the percentage of effectiveness implies.
L40: raising effectiveness from 80 to 90 % due to addition of another insecticide doesn’t prove elevated efficiency, potentiation etc.
L92: from successfully molting = from successful molting
L137: of the L-strain = as the L-strain
L145: in each insecticide concentration = in insecticide solution of a given concentration
L273: two tested instar larvae = two tested instars
L287-288 and elsewhere: rephrase to avoid repetition of terms, such as “larval duration”
L297: “reduction” doesn’t need a plural form
L305: fecundity decreased from – rephrase for clarity
L317 and possibly elsewhere: do not forget to use italics for Latin names
L325: Our result revealed the efficiency – rephrase for simplicity
L359 and further: what does it mean “lowest/highest significant activity”?
L447: the abbreviations “F-strain” and “L-strain” are not used in the illustration. The term “strain” is mentioned too often. Rephrase for simplicity
L531: activation of what?
L555: Chemical insecticides continue = Chemical insecticides application continues
L556-557: to control Egyptian crops – what do you mean? Pests are t obe controlled while crops are to be protected
Author Response
Thank you so much for your insightful feedback; we truly appreciate the time and effort you’ve dedicated to helping us. We are especially grateful, as you not only provide comments but also guide us toward effective solutions. We have made all the suggested revisions and are hopeful that these changes align with your expectations.
- First of all, why control is absent from all data sheets and graphs?
Response: We used a control group for all experiments in this manuscript.
*For the bioassay experiment, we used distilled water as the control group to confirm that no mortality was observed in the tested insects under the distilled water. So, no need to use the corrected mortality equation.
*For biological parameters experiments, we already used the control and inserted it in Table 2 in the first row.
*For field experiments, we used the control results in the Henderson-Tilton equation to get the % mortality
Corrected Mortality= (M Treated – M Control​)/(100−M Control​)​×100
Where:
Mâ‚“ = Mortality rate in each group (treated or control).
- M Treated​ is the mortality in the treated group.
- M Control​ is the mortality in the control group (non-treated).
*For the enzyme assay, we determined the enzyme activities in field and laboratory strains to compare the enzyme amount or activity in each tested strain under the tested insecticides and their mixtures.
- Toxicology …. on Spodoptera – unclear wording.
Response: rephrased it.
- L32: aim .. investigates – a wrong choice of subject vs predicate
Response: modified it.
- L38: it is not clear what the percentage of effectiveness implies.
Response: Sorry for this mistake. Modified it.
- L40: raising effectiveness from 80 to 90 % due to addition of another insecticide doesn’t prove elevated efficiency, potentiation etc.
Response: modified it.
- L92: from successfully molting = from successful molting
Response: Done in line 98.
- L137: of the L-strain = as the L-strain
Response: modified in line 143.
- L145: in each insecticide concentration = in insecticide solution of a given concentration
Response: Done in line 151.
- L273: two tested instar larvae = two tested instars
Response: Done in line 277.
- L287-288 and elsewhere: rephrase to avoid repetition of terms, such as “larval duration”
Response: Done in lines 291 and 293.
- L297: “reduction” doesn’t need a plural form
Response: modified in line 302.
- L305: fecundity decreased from – rephrase for clarity
Response: rephrased in line 310.
- L317 and possibly elsewhere: do not forget to use italics for Latin names
Response: Done in line 323.
- L325: Our result revealed the efficiency – rephrase for simplicity
Response: modified in line 331.
- L359 and further: what does it mean “lowest/highest significant activity”?
Response: In this version, I replaced "highest" and "lowest" with terms like greatest, minimal, most pronounced, strongest, notable, and marked, to avoid repetition and offer more variety in language.
- L447: the abbreviations “F-strain” and “L-strain” are not used in the illustration. The term “strain” is mentioned too often. Rephrase for simplicity
Response: rephrased.
- L531: activation of what?
Response: clarify it in line 600.
- L555: Chemical insecticides continue = Chemical insecticides application continues
Response: replaced in line 625.
- L556-557: to control Egyptian crops – what do you mean? Pests are to be controlled while crops are to be protected
Response: Done in line 627.